# Genome-Wide Identification and Analysis of Fruit Expression Patterns of the *TCP* Gene Family in Three Genera of Juglandaceae

**DOI:** 10.3390/biology14111529

**Published:** 2025-10-30

**Authors:** Shengjie Sun, Xiaodong Wu, Jiaole Liu, Yinlong Zhang, Rui Shi, Dan Li

**Affiliations:** 1Key Laboratory for Forest Resources Conservation and Utilization in the Southwest Mountains of China, Southwest Forestry University, Ministry of Education, Kunming 650224, China; sunshengjie@swfu.edu.cn (S.S.); 13618757367@swfu.edu.cn (J.L.); zuiyanli@swfu.edu.cn (Y.Z.); 2Yunnan Provincial Key Laboratory for Conservation and Utilization of In-Forest Resource, Southwest Forestry University, Kunming 650224, China; wuxiaodong1547@swfu.edu.cn

**Keywords:** *TCP* gene family, *Carya illinoinensis*, *Annamocarya sinensis*, *Juglans regia*, bioinformatics analysis, fruit, expression analysis

## Abstract

**Simple Summary:**

Nut-producing trees such as pecan, beaked walnut, and walnut are valued for their high nutritional quality, but the genetic basis of their fruit development is still not well understood. Genes known as *TCP* genes play key roles in controlling plant growth, development, and responses to the environment. In this study, we carefully searched the entire genomes of pecan, beaked walnut, and walnut and identified the members of the *TCP* gene family in each species. We then examined their characteristics, positions on chromosomes, evolutionary relationships, and activity during fruit growth. Our results showed that many *TCP* genes are active in fruits, especially a group called the CIN type, which may be particularly important for shaping fruit structure and supporting fruit maturation. We also confirmed that some specific genes in each species are strongly expressed when the fruits are developing, suggesting they could be crucial for determining fruit quality. This research improves our understanding of how nut fruits develop at the genetic level and provides useful information that may support future breeding efforts to produce nut varieties with better yield, quality, and resilience to environmental stress.

**Abstract:**

The *TCP* gene family plays essential roles in plant growth, development, and stress responses, yet their evolutionary dynamics and functional characteristics remain poorly understood in Juglandaceae species. Here, we aimed to systematically identify, classify, and characterize *TCP* genes across three nut-producing Juglandaceae species—*Carya illinoinensis*, *Annamocarya sinensis*, and *Juglans regia*—to elucidate their evolutionary relationships and potential functions in fruit development. We identified 44, 35, and 36 *TCP* genes in *C. illinoinensis*, *A. sinensis*, and *J. regia*, respectively, and classified them into three subfamilies (PCF, CIN, and CYC/TB1). Physicochemical property analysis revealed that most proteins were hydrophilic but relatively unstable. Conserved motif and gene structure analyses showed strong similarity among closely related members, while promoter regions were enriched with cis-acting elements associated with development, hormone signaling, and stress responses. Chromosomal mapping demonstrated an uneven distribution of *TCP* genes, with frequent clustering, and synteny analysis indicated strong conservation and gene duplication within and across species. Transcriptome profiling revealed that approximately half of the *TCP* genes were expressed in fruit tissues, with CIN subfamily members showing preferential expression. qRT-PCR validation further highlighted *AsTCP23*, *CiTCP14*, and *JrTCP09* as highly expressed during fruit development, suggesting potential regulatory roles in fruit maturation. These findings provide new insights into the evolutionary patterns and functional divergence of *TCP* genes in Juglandaceae and establish a valuable foundation for future studies on fruit development and genetic improvement. Collectively, these findings advance our understanding of *TCP* gene evolution and provide potential molecular targets for improving fruit development and nut quality in Juglandaceae crops.

## 1. Introduction

Transcription factors (TFs) bind specifically to the promoter regions of eukaryotic genes, playing a pivotal role in plant growth, development and responses to environmental stresses [1,2,3]. The *TCP* gene family represents a class of transcription factors unique to plants. Its name derives from the first members identified, including Teosinte Branched1 (TB1) from maize (*Zea mays*), Cycloidea (CYC) from snapdragon (*Antirrhinum*), and Proliferating Cell Nuclear Antigen Factors 1 and 2 (PCF1 and PCF2) from rice (*Oryza sativa*) [4,5,6]. Members of the *TCP* gene family possess a highly conserved basic helix–loop–helix (bHLH) structure, 59 amino acids in length, located at the N-terminus [7]. This conserved domain, referred to as the TCP domain, is associated with DNA binding, protein–protein interactions, and nuclear localization [8]. Based on sequence characteristics and phylogenetic relationships of the TCP domain, the family can be divided into two major subclasses: Class I and Class II. Class I *TCP* gene family are further grouped into the CIN and CYC/TB1 clades, whereas Class II members are referred to as the PCF clade. The most prominent distinction between these two subclasses lies in the basic region of the TCP domain: Class I proteins contain four additional amino acids compared with those of Class II [8,9]. Notably, the CYC/TB1 clade is unique to angiosperms and is characterized by the presence of an arginine-rich motif (R domain) as well as the glutamic acid–cysteine–glutamic acid (ECE) motif, both of which contribute to protein–protein interactions [10,11].

Fruit development is generally considered a sequential process comprising three major stages: initiation, growth, and maturation [12]. Members of the *TCP* gene family exert considerable influence on these developmental processes and are also involved in regulating diverse hormone biosynthetic pathways. For example, in *Arabidopsis thaliana*, *TCP* genes modulate the activation of seed embryonic growth potential [13]. In grape (*Vitis vinifera*), the expression of most *TCP* genes is suppressed by drought and waterlogging stresses, thereby affecting early fruit development [14]. In sweet cherry (*Prunus avium*), *TCP* gene family members regulate light perception to control the synthesis of secondary metabolites, which strongly contributes to fruit quality [15]. In cruciferous species, *TCP* genes play critical roles in determining fruit dehiscence [16]. *TCP* gene family members have been identified in many plant species. Therefore, a systematic characterization of the *TCP* gene family in Juglandaceae is expected to provide valuable resources for further studies in this family.

The Juglandaceae family comprises approximately 10 genera and 60 species worldwide, with about 7 genera and 25 species distributed in China [17,18]. Many members of this family are of significant economic and ecological value, being recognized as important nut, medicinal, and timber trees [19]. Their fruits serve as valuable sources of food, oil, and traditional medicine [20]. In particular, species from the genera *Carya*, *Juglans*, and *Annamocarya* produce highly nutritious nuts [21,22,23]. The pecan (*Carya illinoinensis*) is notable for its large, thin-shelled nuts that are easy to crack, making it an important dual-purpose species for both oil and food production [24,25]. The walnut (*Juglans regia*) is a widely consumed dry fruit rich in nutrients; moreover, its roots, young shoots, leaves, flowers, seed kernels, and fruit septa are all used in traditional medicine [26]. The beaked walnut (*Annamocarya sinensis*, also referred to as *Carya sinensis*) is a relict species from the ancient tropics of the Tertiary period’s ancient tropics of the Juglandaceae [27,28]. Its kernels are rich in amino acids and fatty acids, with high oil content [23]. With the rapid development of sequencing technologies, chromosome-level genome assemblies of *C. illinoinensis*, *J. regia*, and *A. sinensis* have recently become available [29,30]. While the *TCP* gene family has been analyzed in *C. illinoinensis*, no comparative studies have addressed its evolutionary relationships within Juglandaceae or its expression patterns during fruit development [31]. Therefore, conducting a genome-wide identification of *TCP* genes and analyzing their expression across fruits of these three representative species will provide valuable insights into their evolution and functional roles.

In this study, we performed a genome-wide identification of the *TCP* gene family in three Juglandaceae species with high nutritional value: *Carya illinoinensis*, *Annamocarya sinensis*, and *Juglans regia*. The main objectives of this study were to (1) comprehensively identify *TCP* genes in the three species, (2) analyze their phylogenetic relationships, conserved motifs, sequence alignment, chromosomal distribution, synteny, and cis-acting regulatory elements, and (3) explore their expression patterns in fruit tissues to reveal potential roles in fruit growth and development. These results provide a molecular basis for understanding the regulatory roles of *TCP* genes in fruit growth and development within Juglandaceae.

## 2. Materials and Methods

### 2.1. Materials

Samples of *A*. *sinensis* were collected on 3 October 2024, from a wild population near Fuli Village, Funing County, Wenshan Zhuang, and Miao Autonomous Prefecture, Yunnan Province, China. Healthy trees without visible signs of pests or diseases were selected, and nearly mature fruits with lengths > 5 cm and diameters > 4 cm were harvested. Fruits of *C*. *illinoinensis* and *J*. *regia* were collected on 20 September 2024, and 9 July 2025, respectively, from the arboretum of Southwest Forestry University, Kunming, Yunnan Province. For pecan, nearly mature fruits with lengths > 3 cm and diameters > 1.5 cm were chosen, whereas walnut fruits with diameters > 4 cm were selected. All collected fruits were rinsed with sterile water, immediately frozen in liquid nitrogen, and stored at −80 °C until use. The plant species were identified by Professor Xueli Zhao (Southwest Forestry University, China). Three biological replicates were included for each experiment.

### 2.2. Methods

#### 2.2.1. Identification of the *TCP* Gene Family

The genome sequences and annotation files of *C*. *illinoinensis* (BioProject: PRJNA680555) and *J*. *regia* (BioProject: PRJNA291087) were obtained from the Genome Warehouse (GWH, https://ngdc.cncb.ac.cn/gwh/, accessed on 15 May 2025), while those of *A*. *sinensis* were retrieved from the Juglandaceae Genome Projects (https://cmb.bnu.edu.cn/juglans/, accessed on 15 May 2025). The Hidden Markov Model profile of the TCP domain (PF03634) [8,32] was downloaded from the Pfam database (http://pfam.xfam.org/, accessed on 15 May 2025) and used to identify candidate proteins with TBtools-II v2.154 [33]. TCP sequences of *Arabidopsis thaliana* were acquired from TAIR (https://www.arabidopsis.org/, accessed on 15 May 2025) and applied as queries in TBtools-II BLAST searches against the genomes of *C*. *illinoinensis*, *J*. *regia*, and *A. sinensis*. Redundant sequences were removed, and the remaining candidates were further validated using the Batch Web CD-Search Tool (https://www.ncbi.nlm.nih.gov/Structure/bwrpsb/bwrpsb.cgi/, accessed on 15 May 2025) [34]. Sequences lacking intact TCP domains were excluded, resulting in the final set of *TCP* gene family members in the three Juglandaceae species.

#### 2.2.2. Physicochemical Property Analysis of TCP Proteins

The physicochemical properties of TCP proteins were analyzed using the Protein Parameter Calculator in TBtools. Transmembrane domains were predicted with the online tool TMHMM v2.0 (https://services.healthtech.dtu.dk/services/TMHMM-2.0/, accessed on 16 May 2025).

#### 2.2.3. Phylogenetic Tree Construction and Visualization

Phylogenetic analysis of TCP proteins was performed in MEGA11 using the neighbor-joining (NJ) method with 1000 bootstrap replicates to assess node reliability. The resulting tree was subsequently refined and visualized using Evolview (https://www.evolgenius.info/evolview/, accessed on 16 May 2025) [35].

#### 2.2.4. Conserved Motif, Gene Structure and Promoter Analysis

Conserved motifs of TCP proteins from the three species were identified using MEME (https://web.mit.edu/meme/current/share/doc/meme.html, accessed on 16 May 2025)), with the maximum motif number set to 10. Promoter sequences (2000 bp upstream of the start codon) were extracted with TBtools-II, and cis-acting regulatory elements were annotated using PlantCARE (https://bioinformatics.psb.ugent.be/webtools/plantcare/html/, accessed on 16 May 2025)). Gene structures, conserved motifs, and cis-elements were visualized with the Gene Structure View (Advanced) module in TBtools-II

#### 2.2.5. Chromosomal Localization, Gene Duplication and Syntenic Analysis

Gene duplication events and syntenic relationships of *TCP* genes within and among *C. illinoinensis*, *J. regia*, and *A. sinensis* were identified using the MCScanX module in TBtools-Ⅱ. Both tandem and segmental duplications were examined to explore the evolutionary dynamics of the *TCP* gene family. The results were visualized using Multiple Synteny Plot and Advanced Circos tools.

#### 2.2.6. Expression Pattern Analysis of *TCP* Genes

Fruit transcriptome datasets of *C. illinoinensis* (PRJNA1180220), *J. regia* (PRJNA781571), and *A. sinensis* (PRJNA1262124) were retrieved from the NCBI SRA database using the SRA Toolkit. Clean reads were mapped to the respective reference genomes, and transcript abundance was quantified as TPM values using the RNA-seq module in TBtools-II. Heatmaps of *CiTCPs*, *AsTCPs*, and *JrTCPs* expression profiles were generated with the HeatMap module in TBtools-II.

#### 2.2.7. RNA Extraction and qRT-PCR Analysis

Total RNA was isolated using the RNA extraction kit (Vazyme, Nanjing, China), and first-strand cDNA was synthesized with a reverse transcription kit (Applied Biological Materials, Richmond, BC, Canada) according to the manufacturers’ protocols. In order to investigate the expression patterns of *TCP* genes during fruit development in Juglandaceae species, two *TCP* genes with the highest TPM values and one CYC/TB1 subfamily gene with no detectable expression were selected from each species. *Actin* genes of *A. sinensis*, *J. regia*, and *C. illinoinensis* served as internal references. Gene-specific primers were designed using Primer Premier 5 (detailed in Appendix A). qRT-PCR was performed with the qRT-PCR kit (Applied Biological Materials, Canada) on a LightCycler^®^ 96 Instrument (Roche, Basel, Switzerland). Ct values were obtained, and relative gene expression levels were calculated using the 2^−ΔΔCt^ method. Each reaction was conducted with three biological replicates and three technical replicates.

## 3. Results

### 3.1. Identification and Physicochemical Properties of TCP Gene Family

A total of 115 *TCP* genes were identified across the three Juglandaceae species, including 44 in *C. illinoinensis* (*CiTCPs*), 36 in *J. regia* (*JrTCPs*), and 35 in *A. sinensis* (*AsTCPs*). The genes were named sequentially according to their chromosomal positions, ranging from *CiTCP01*–*CiTCP44*, *JrTCP01*–*JrTCP36*, and *AsTCP01*–*AsTCP35*. The predicted physicochemical properties are summarized in Table 1. The TCP proteins varied from 133 to 727 amino acids in length, with molecular weights ranging between 14.0 and 79.4 kDa. Among them, 70 proteins had molecular weights greater than 40 kDa, indicating that most members are of medium size. The theoretical isoelectric points (pI) ranged from 5.5 to 10.4. The instability index was between 38.3 and 77.5, with only three proteins classified as stable (index < 40) and 82 proteins showing values > 50, suggesting that most TCP proteins are relatively unstable. The average hydropathy values were all negative, reflecting the hydrophilic nature of these proteins. Prediction of transmembrane domains revealed that only *JrTCP01*, *JrTCP09*, and *AsTCP18* are likely transmembrane proteins (Appendix A), while no transmembrane proteins were detected among *CiTCP* members.

### 3.2. Phylogenetic Analysis of the TCP Gene Family

Based on the phylogenetic relationships, the *TCP* gene family members were classified into three subfamilies: PCF, CYC/TB1, and CIN (Figure 1). In the PCF and CYC/TB1 subfamilies, *C. illinoinensis* contained 19 and 5 members, *A. sinensis* contained 18 and 6 members, and *J. regia* contained 18 and 7 members, respectively. The distribution of genes across these two subfamilies was relatively balanced among the three species, with members from each species appearing in almost every clade, suggesting limited divergence. In contrast, clear differences were observed in the CIN subfamily. *C. illinoinensis* contained 20 CIN members, nearly twice the number found in *A. sinensis* and *J. regia*, which harbored only 11 members each. Notably, *CiTCP19*–*CiTCP24* and *CiTCP36*–*CiTCP39* formed a closely related clade, indicating that *C. illinoinensis* may have undergone a recent gene expansion event. These results highlight both conserved and divergent evolutionary patterns of *TCP* genes in Juglandaceae species.

### 3.3. Conserved Motifs, Gene Structure, and Promoter Analysis of TCP Gene Family

Analysis of conserved motifs and cis-acting elements revealed that *TCP* genes (Figure 2a) with close phylogenetic relationships generally exhibited similar types and numbers of motifs (Figure 2b) and regulatory elements (Figure 2c). In total, 10 conserved motifs (Motif 1–Motif 10) (Appendix A) were identified among the TCP proteins from the three Juglandaceae species, with lengths ranging from 21 to 50 amino acids. Motif 1 was present in all 115 TCP members. Motifs 4, 6, 7, and 8 were restricted to the CIN subfamily (except Motif 4, which was also found in JrTCP19 of the PCF subfamily). Motif 2 was specific to the PCF subfamily, whereas Motifs 5 and 10 were present only in the CIN and PCF subfamilies. Motif 3 was detected in both the CIN and CYC/TB1 subfamilies. The exon–intron structures of *TCP* genes fell into three categories, containing one, two, or five exons (Figure 2d). Most genes (approximately 85%) possessed a single exon. In the CIN subfamily, all members except *JrTCP09* and *AsTCP18* had only one exon. Within the CYC/TB1 subfamily, eight genes (approximately 47%) had two exons, while the remainder contained one. In the PCF subfamily, four members harbored two exons, and three genes with five exons were exclusively found in this subfamily. Among the species, *C. illinoinensis* contained the highest number of two-exon genes (eight, distributed in both the CYC/TB1 and PCF subfamilies). *A. sinensis* and *J. regia* each contained five two-exon genes, distributed across all three subfamilies in the former, but only in the CIN and CYC/TB1 subfamilies in the latter. Each species also had a single *TCP* gene with five exons. The distribution of non-coding regions was diverse. Cis-acting element prediction within the 2000 bp upstream promoter regions identified 2819 elements in total: 1091 in *C. illinoinensis*, 861 in *A. sinensis*, and 867 in *J. regia*. These elements were classified into four major categories: light-responsive, hormone-responsive, stress-responsive, and growth/development-related. Light-responsive elements were the most abundant (approximately 12.2 per gene), followed by hormone-responsive (approximately 7.5 per gene), stress-responsive (approximately 3.7 per gene), and growth/development-related elements (approximately 0.9 per gene). Notably, 2483 cis-elements (approximately 21.6 per gene) were associated with fruit development and regulation.

### 3.4. Chromosomal Localization and Gene Collinearity Analysis of TCP Genes

Chromosomal mapping revealed that 44, 35, and 36 *TCP* genes were identified in *C. illinoinensis*, *A. sinensis*, and *J. regia*, respectively, with an uneven distribution across the chromosomes (Figure 3). These genes were mainly located in gene-rich regions, while no *TCP* members were detected on CiChr07, CiChr08, CiChr13, AtChr07, AtChr08, AtChr13, JrChr08, JrChr11, and JrChr14. Intra-chromosomal duplication events were detected in *C. illinoinensis* on CiChr15 (*CiTCP41* and *CiTCP42*) and CiChr16 (*CiTCP43* and *CiTCP44*), as well as in *A. sinensis* on AsChr16 (*AsTCP34* and *AsTCP35*). In addition, 31, 30, and 26 inter-chromosomal duplication events were identified in *C. illinoinensis*, *A. sinensis*, and *J. regia*, respectively.

Gene collinearity analysis across species (Figure 4) indicated no significant syntenic relationship between *Arabidopsis thaliana* and Juglandaceae species. In contrast, extensive synteny was observed among the three Juglandaceae species. A total of 87 syntenic gene pairs were identified between *J. regia* and *C. illinoinensis*, while 98 gene pairs were detected between *C. illinoinensis* and *A. sinensis*.

### 3.5. Expression Patterns of TCP Genes in Fruit

Analysis of fruit transcriptomes revealed that 22 out of 44 *TCP* genes in *C. illinoinensis* were expressed in the fruit, with *CiTCP28* and *CiTCP14* exhibiting markedly higher expression levels than other family members (Figure 5a). In *A. sinensis*, 22 of 35 *TCP* genes were expressed in the fruit, among which *AsTCP23* and *AsTCP19* showed the highest transcript abundance (Figure 5b). For *J. regia*, 19 of 36 *TCP* genes were expressed in fruit, with *JrTCP10* and *JrTCP09* displaying prominent expression levels (Figure 5c). Across the three Juglandaceae species, *TCP* genes with high fruit expression were primarily concentrated in the CIN subfamily, while no expression was detected in the CYC/TB1 subfamily (Figure 5d).

To validate the transcriptome results, representative *TCP* genes in fruit were analyzed using qRT-PCR. The expression trends observed by qRT-PCR were consistent with the transcriptome data, confirming that *CiTCP28*, *CiTCP14*, *AsTCP23*, *AsTCP19*, *JrTCP10*, and *JrTCP09* are highly transcriptionally active during fruit development (Figure 6). Further quantitative analysis indicated species-specific differences in relative expression levels. In *A. sinensis*, *AsTCP23* was expressed significantly higher than *AsTCP19* (Figure 6a). In *C. illinoinensis*, *CiTCP14* showed higher expression than *CiTCP28* (Figure 6b). In *J. regia*, *JrTCP09* expression exceeded that of *JrTCP10* (Figure 6c), suggesting that these genes may play key regulatory roles during the late stages of fruit maturation.

## 4. Discussion

*TCP* gene family has been identified in several plant species, including 24 members in *Arabidopsis thaliana* [36], 21 in rice (*Oryza sativa*) [37], 15 in grape (*Vitis vinifera*) [38], and 36 in poplar (*Populus tomentosa*) [39]. In the present study, 44, 35, and 36 TCP genes were identified in *C. illinoinensis*, *A. sinensis*, and *J. regia*, respectively, indicating that these Juglandaceae species generally harbor a larger *TCP* gene family. Previous studies have suggested that several Juglans species underwent whole-genome duplication (WGD) and polyploidization events approximately 24 million years ago [40,41]. Such events may explain the relatively high number of *TCP* genes in Juglandaceae, and it is plausible that Carya species, including *C. illinoinensis* and *A. sinensis*, also experienced similar genome duplications. Physicochemical property analysis showed that most TCP proteins were highly hydrophilic and generally unstable [42]. Their amino acid lengths ranged from 133 to 727 residues, and theoretical isoelectric points (pI) varied between 5.5 and 10.4, reflecting potential functional diversity within the family. Notably, TCP proteins in *A. sinensis* displayed the greatest variability: *AsTCP18* was the longest (727 amino acids) and one of the few TCP proteins potentially containing transmembrane regions, whereas *AsTCP31* was the shortest (133 amino acids). This diversity may reflect the relict nature of *A. sinensis*, which has retained a broader spectrum of *TCP* gene variants. Phylogenetic analysis grouped the *TCP* genes of all three species into three subfamilies: PCF, CYC/TB1, and CIN. The CYC/TB1 subfamily contained the fewest members, consistent with TCP classifications reported in other plant species (Figure 1) [43,44,45,46].

*TCP* genes play critical roles in plant growth, development, and responses to abiotic stresses. Sequence analysis of *TCP* gene family members in the three Juglandaceae species revealed that genes with close phylogenetic relationships generally shared similar conserved motifs and cis-regulatory elements. Across the three species, a total of 115 *TCP* genes were identified, and 10 distinct conserved motifs were detected. The type, arrangement, and number of motifs varied substantially among different subfamilies. Notably, Motif 1 was present in all *TCP* genes, likely corresponding to the core TCP domain characteristic of this family. Most *TCP* genes (approximately 85%) contained a single exon, while a few had two to five exons. This exon structure diversity suggests that *TCP* genes have undergone varying degrees of splicing modifications and rearrangements during evolution, which may contribute to their functional diversification [47]. Differences in motif composition and exon organization were also observed between different evolutionary branches of each subfamily. However, genes from the three species within the same evolutionary branch often clustered together, with each cluster containing one gene from each species. Within these clusters, motif composition and exon organization were highly conserved, highlighting that structural differences among subfamilies, branches, or groups may underlie functional variation within the *TCP* gene family. Promoter analysis identified four major categories among the 115 *TCP* genes: light-responsive elements, hormone-responsive elements, stress-responsive elements, and growth and development-related elements. These findings suggest that *TCP* genes may play key roles in organ development, hormone signaling, and abiotic stress responses in Juglandaceae species.

Chromosomal mapping of *TCP* genes was conducted for *C. illinoinensis*, *A. sinensis*, and *J. regia*, and both intraspecies and interspecies gene collinearity were systematically analyzed. The results revealed that *TCP* genes are unevenly distributed along chromosomes, often forming clusters. This distribution pattern is highly similar to that observed in model plants such as *Arabidopsis thaliana*, maize, and tomato, suggesting that chromatin remodeling and segmental rearrangements may jointly shape the chromosomal organization of *TCP* genes during plant genome evolution [9]. Analysis of interchromosomal duplication events showed a relatively high rate of gene duplication among the three Juglandaceae species. Such duplications not only expand the overall genetic repertoire of the *TCP* gene family but also provide opportunities for functional diversification of redundant genes over evolutionary time. Interspecies synteny analysis further demonstrated strong conservation of *TCP* gene structures within Juglandaceae. This high degree of collinearity indicates that *TCP* genes have maintained structural and regulatory conservation throughout evolution, likely contributing to consistent regulatory networks and biological functions across species [42].

Transcriptome analysis revealed distinct expression profiles of *TCP* gene family members in the fruit tissues of the three Juglandaceae species. Approximately half of the genes displayed measurable expression in fruits, suggesting a degree of tissue specificity during fruit development. This observation is consistent with the organ-preferential expression patterns previously reported for the *TCP* gene family [15,48]. Genes with relatively high transcript abundance in fruits were largely restricted to the CIN subfamily, whereas members of the CYC/TB1 subfamily showed little or no expression. This contrast reflects the functional divergence of the two subfamilies. The CIN subfamily has been implicated in leaf development, cell expansion, and fruit organ morphogenesis in several plant species, while CYC/TB1 subfamily mainly regulate apical dominance, branching, and floral symmetry, processes that are less directly linked to fruit development. Together, these findings suggest that CIN subfamily may play a predominant role in fruit morphogenesis, cell fate determination, and maturation in Juglandaceae plants [49,50]. qRT-PCR assays not only validated the transcriptome data but also highlighted several genes with markedly increased expression during the late stages of fruit development, such as *AsTCP23*, *CiTCP14*, and *JrTCP09*. The elevated expression of these genes may reflect their involvement in cellular restructuring, nutrient accumulation, and enhanced metabolic activity during fruit maturation, thereby contributing to the formation of final fruit quality traits. Nevertheless, this study is mainly based on bioinformatics analyses, with experimental validation limited to qRT-PCR. Future experiments are planned to expand these findings and provide more comprehensive insights into the molecular mechanisms underlying fruit development and quality formation.

## 5. Conclusions

In this study, we identified 115 *TCP* gene family members from three Juglandaceae species and systematically analyzed their sequence characteristics, evolutionary relationships, and expression profiles. These genes were assigned to the PCF, CIN, and CYC/TB1 subfamilies, reflecting both strong conservation and clear evidence of functional divergence. Expression analyses showed that roughly half of the family members were active in fruit tissues, with the majority of highly expressed genes clustering within the CIN subfamily. This pattern points to a central role of CIN subfamily in regulating fruit development. qRT-PCR assays further confirmed the transcriptome findings and highlighted *AsTCP23*, *CiTCP14*, and *JrTCP09* as representative genes with pronounced expression increases during fruit maturation. Their expression trends suggest possible involvement in late-stage developmental transitions, including tissue differentiation and metabolic activity associated with fruit quality. Overall, this study provides new insights into the evolutionary history and functional specialization of *TCP* genes in Juglandaceae. Beyond advancing basic understanding, these findings identify promising *TCP* candidates for molecular breeding aimed at enhancing fruit development, yield, and quality in nut crops.

## Figures and Tables

**Figure 1 biology-14-01529-f001:**
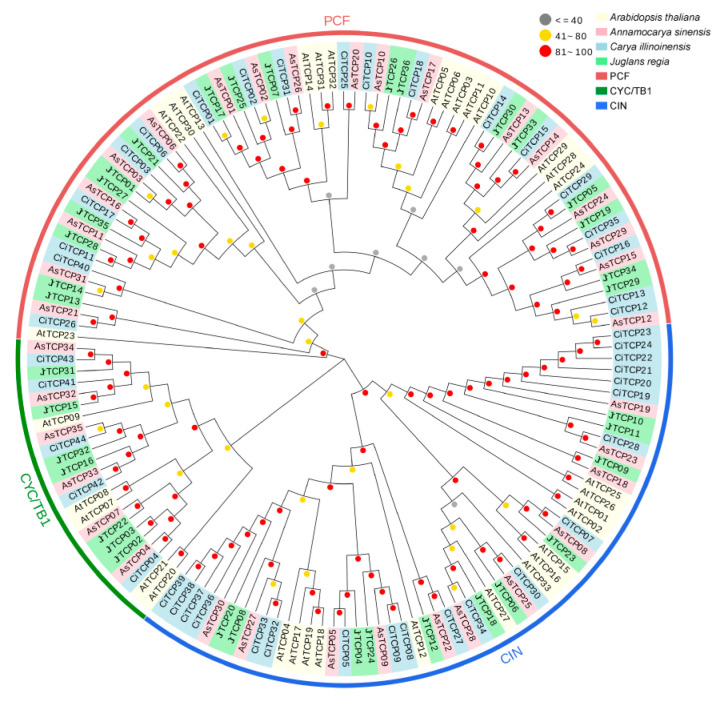
Phylogenetic tree of *TCP* gene family in *C. illinoinensis*, *A. sinensis*, *J. regia* and *A. thaliana*.

**Figure 2 biology-14-01529-f002:**
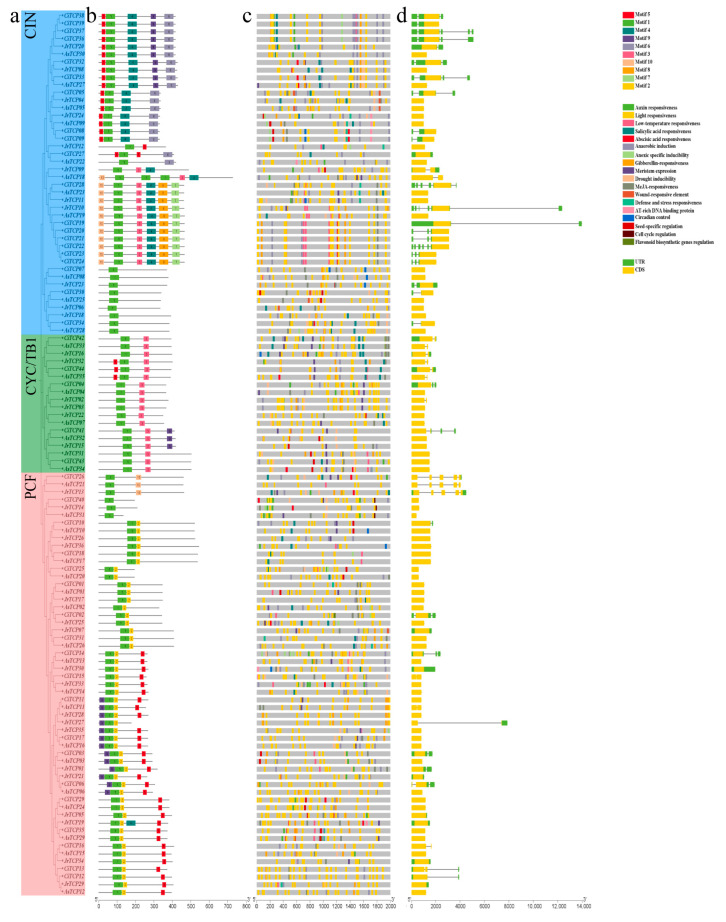
Phylogenetic relationships, conserved motifs, distribution of cis-acting element in the promoters and gene structure of the *TCP* gene family members in *C. illinoinensis*, *A. sinensis*, and *J. regia*. (**a**) Phylogenetic analysis; (**b**) conserved motifs; (**c**) distribution of promoter cis-acting elements; and (**d**) gene structure. In the promoter cis-acting elements section, light gray bands indicate regions with no predicted cis-acting elements, while different color blocks represent different cis-acting elements.

**Figure 3 biology-14-01529-f003:**
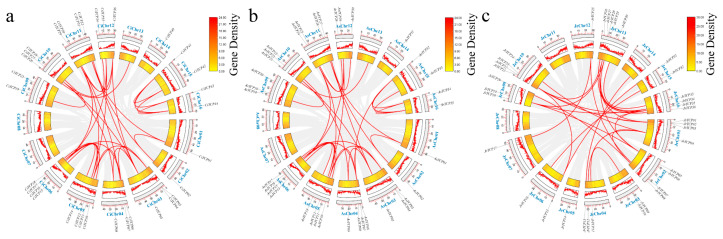
Collinearity analysis and chromosomal position of *TCP* gene family in *C. illinoinensis*, *A. sinensis* and *J. regia*. Note: (**a**) *A. sinensis*; (**b**) *C. illinoinensis*; (**c**) *J. regia*. From the outer circle to the inner circle, gene name, chromosome name, chromosome length, gene density curve, gene density heatmap and collinearity regions are indicated. The gray line in the background of the circle represents the gene pairs of all fragment duplications in the genome, and the red line represents the *TCP* gene pairs with fragment duplications.

**Figure 4 biology-14-01529-f004:**
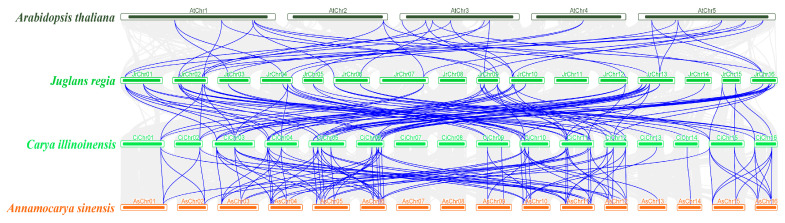
Collinearity analysis of *TCP* gene family among *A. thaliana*, *J. regia*, *C. illinoinensis* and *A. sinensis*. Note: Chr refers to chromosome. The putative collinear genes in *A. thaliana*, *J. regia*, *C. illinoinensis* and *A. sinensis* are marked in gray, while the syntonic *TCP* gene pairs are marked in blue.

**Figure 5 biology-14-01529-f005:**
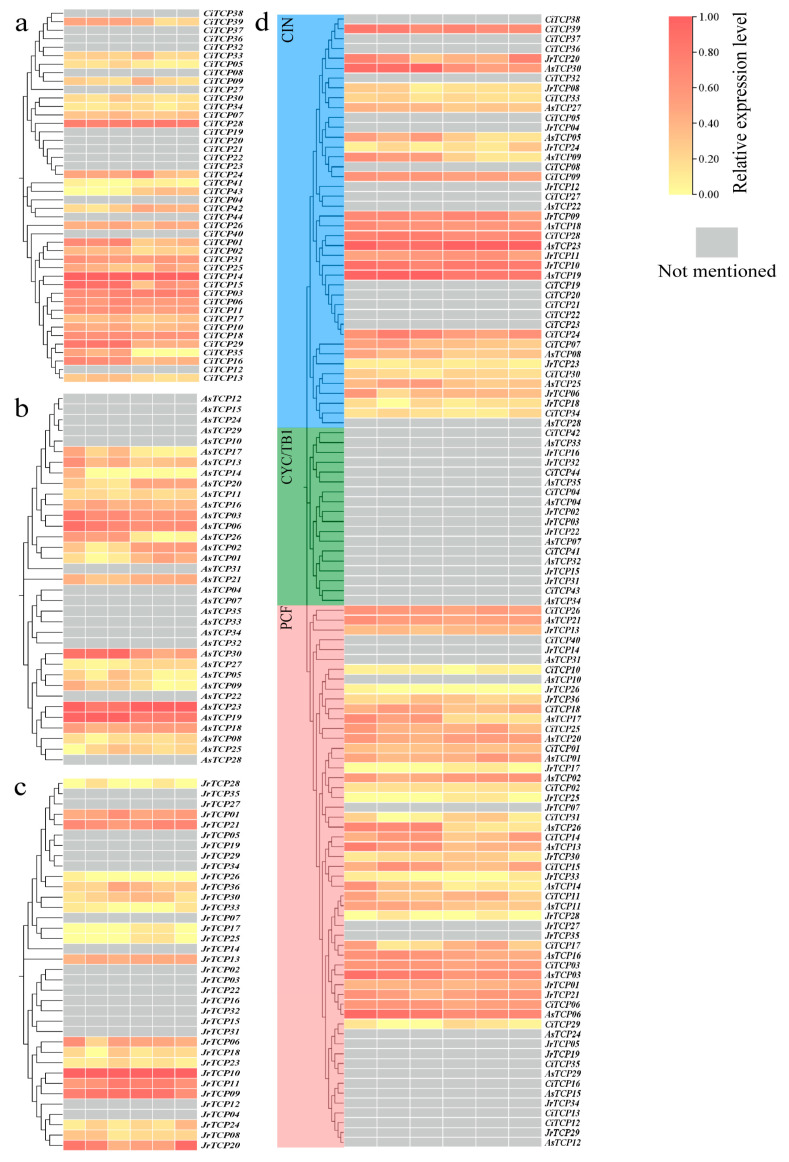
Expression Analysis of *TCP* Gene Family Members in Fruits of three Juglandaceae species. Note: (**a**) *C. illinoinensis*; (**b**) *A. sinensis*; (**c**) *J. regia*; (**d**) Summary of three species. First, the transcriptome data were normalized using the log_2_(TPM) method, followed by zero-to-one normalization of each row of data. In the heatmap, red indicates higher expression levels, yellow indicates lower expression levels, and gray represents no expression.

**Figure 6 biology-14-01529-f006:**
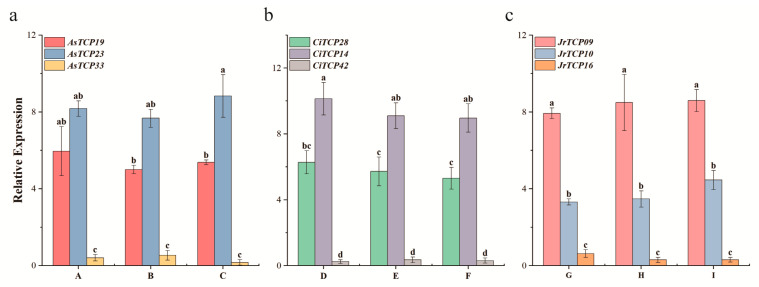
Expressions of *TCP* gene family in fruit development of three Juglandaceae species. Note: (**a**) *A. sinensis* (A, B, and C represent three biological replicates); (**b**) *C. illinoinensis* (D, E, and F represent three biological replicates); (**c**) *J. regia* (G, H, and I represent three biological replicates). Data are presented as the mean ± SD of three technical replicates. Different letters indicate significant difference at *p* ≤ 0.05 level.

**Table 1 biology-14-01529-t001:** Analysis of physicochemical properties of *TCP* gene family members in Juglandaceae plants.

Species	Gene Number	Number of Amino Acid	Molecular Weight (kDa)	Theoretical pI	Instability Index	Grand Average of Hydropathicity	Transmembrane- Domain Number
*Carya illinoinensis*	44	537~193	56.2~20.9	9.8~5.9	71.8~38.3	−0.17~−0.87	0
*Juglans regia*	36	543~177	57.0~18.8	10.1~5.5	67.0~38.5	−0.20~−0.88	0~2
*Annamocarya sinensis*	35	727~133	79.4~14.0	10.4~5.8	77.5~39.7	−0.16~−0.86	0~1
Total	115	727~133	79.4~14.0	10.4~5.5	77.5~38.3	−0.16~−0.88	0~3

## Data Availability

Publicly available RNA-Seq datasets were analyzed in this study. These data can be accessed from the NCBI SRA database under BioProject accession numbers PRJNA1180220 (*Carya illinoinensis*), PRJNA781571 (*Juglans regia*), and PRJNA1262124 (*Annamocarya sinensis*). Other data supporting the conclusions of this article are included within the article and its Appendix A.

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
