# Peer review of "Genome-Wide Identification and Analysis of Fruit Expression Patterns of the *TCP* Gene Family in Three Genera of Juglandaceae"

_biology, 2025, doi:10.3390/biology14111529_

Round 1
Reviewer 1 Report
Comments and Suggestions for Authors
Dear Author:
This manuscript conducts a systematic whole-genome identification, evolutionary analysis, and expression pattern study of the TCP gene family in three genera of Juglandaceae (Carya illinoinensis, Annamocarya sinensis, Juglans regia). The research design is reasonable, the methods are comprehensive, the data is detailed, and the results have certain novelty and biological significance. Especially, it provides valuable resources for the functional genomics research of nut crops. It is recommended to accept publication after minor revisions.
The main revision suggestions are as follows:
1. Materials and Methods: The selection of TCP domain (PF03634) requires adding references.
2. Figure 1: It is suggested to add explanations for the identification of different species in the figure, and add "Different colors and symbols represent different crops, and the outer rings of different colors represent different sub-families."
3. Figure 2: It is suggested to upload Motif 1-10 as an attachment and annotate it from left to right with "a-d."
4. It is recommended to add a sentence in the discussion section about the future validation work of these genes. This study mainly involves bioinformatics analysis, and the experimental part only includes qRT-PCR. Even without adding experiments, if one can show the future work plan at the end, it is also worth expecting.
5. In Table 1, "Anamocarya sinensis" has a spelling error, it should be "Annamocarya sinensis."
6. The reference citation format needs to be unified. If there is no doi number, it should not be included.
7. In the Data Availability Statement, it is recommended to add the RNA-Seq database login number (although it can be found in the Materials and Methods).
Date: September 15th, 2025
Author Response
Comments 1: Materials and Methods: The selection of TCP domain (PF03634) requires adding references.
Response 1: Thank you for your valuable suggestion. We have added the relevant references for the selection of the TCP domain (PF03634) in the Materials and Methods section, as recommended. These references have been highlighted in yellow in the revised manuscript (page 4, line 137; page 16, lines 509–512). The following references have been included: "32. Cubas, P.; Lauter, N.; Doebley, J.; Coen, E. The TCP domain: a motif found in proteins regulating plant growth and development. Plant J. 1999, 18, 215-222. 33. Shad, M.A.; Wu, S.; Rao, M.J.; Luo, X.; Huang, X.; Wu, Y.; Zhou, Y.; Wang, L.; Ma, C.; Hu, L. Evolution and Functional Dynamics of TCP Transcription Factor Gene Family in Passion Fruit (Passiflora edulis). Plants 2024, 13, 2568."
Comments 2: Figure 1: It is suggested to add explanations for the identification of different species in the figure, and add "Different colors and symbols represent different crops, and the outer rings of different colors represent different sub-families."
Response 2: Thank you for the helpful comment. We have revised Figure 1 to include the suggested explanations. Different colors and symbols now represent different species, and the outer rings of various colors indicate different subfamilies. This modification has been added to the legend of Figure 1 (page 6, line 221).
Comments 3: Figure 2: It is suggested to upload Motif 1-10 as an attachment and annotate it from left to right with "a-d."
Response 3: Thank you for this useful suggestion. We have annotated the panels in Figure 2 from left to right as “a–d” and uploaded Motifs 1–10 as a supplementary file. We have also revised the figure legend as follows: "Figure 2. Phylogenetic relationships, conserved motifs, distribution of cis-acting elements in the promoters, and gene structure of the TCP gene family members in C. illinoinensis, A. sinensis, and J. regia. (a) Phylogenetic analysis; (b) conserved motifs; (c) distribution of promoter cis-acting elements; and (d) gene structure. Note: in the promoter cis-acting elements section, light grey bands indicate regions with no predicted cis-acting elements, while different color blocks represent different cis-acting elements." We have also updated the Supplementary Materials to include "Figure S2: Motif logo analysis of the TCP gene family." The corresponding revisions are located on page 8, lines 253–258, and page 14, line 415.
Comments 4: It is recommended to add a sentence in the discussion section about the future validation work of these genes. This study mainly involves bioinformatics analysis, and the experimental part only includes qRT-PCR. Even without adding experiments, if one can show the future work plan at the end, it is also worth expecting.
Response 4: We appreciate this valuable suggestion. We have added the following sentence in the Discussion section to address future validation work:" Nevertheless, this study is mainly based on bioinformatics analyses, with experimental validation limited to qRT-PCR. Future experiments are planned to expand these findings and provide more comprehensive insights into the molecular mechanisms underlying fruit development and quality formation." This text has been highlighted in yellow in the revised manuscript (page 13, lines 392–395).
Comment 5: In Table 1, "Anamocarya sinensis" has a spelling error; it should be "Annamocarya sinensis. "
Response 5: Thank you for catching this error. We have corrected the spelling of Annamocarya sinensis in Table 1 accordingly.
Comment 6: The reference citation format needs to be unified. If there is no DOI number, it should not be included.
Response 6: Thank you for pointing this out. We have carefully reviewed and unified all reference citation formats throughout the manuscript.
Comment 7: In the Data Availability Statement, it is recommended to add the RNA-Seq database accession number (although it can be found in the Materials and Methods).
Response 7: Thank you for the suggestion. We have revised the Data Availability Statement to include the RNA-Seq dataset accession numbers as follows: "Publicly available RNA-Seq datasets were analyzed in this study. These data can be accessed from the NCBI SRA database under BioProject accession numbers PRJNA1180220 (Carya illinoinensis), PRJNA781571 (Juglans regia), and PRJNA1262124 (Annamocarya sinensis). Other data supporting the conclusions of this article are included within the article and its Supplementary Materials." This revision appears on page 14, lines 436–440.

Reviewer 2 Report
Comments and Suggestions for Authors
Dear Authors,
I would like to congratulate you on your comprehensive and well-executed study entitled “Genome-Wide Identification and Expression Analysis of TCP Gene Family in Three Juglandaceae Species”. Your manuscript provides a thorough characterization of the TCP gene family in three economically and ecologically important Juglandaceae species, integrating genomic, phylogenetic, structural and transcriptomic analyses, with experimental validation via RT-qPCR. This work represents a significant contribution to the understanding of gene evolution and functional specialization in plants.
The results are robust and clearly presented, highlighting the important role of the CIN subfamily in fruit development. Your study also offers valuable insights that may support future research in plant breeding and biotechnology.
To further strengthen the manuscript, I suggest:
-
Streamline dense technical sections (e.g., lists of motifs, exons, or cis-elements) and consider presenting the data in clear tables either within the main manuscript or as supplementary files to improve readability and clarity.
-
Presenting the research objectives more explicitly.
-
Highlighting potential practical applications or implications for crop improvement in the conclusions.
Overall, your manuscript is scientifically solid and of high relevance and I commend you for your rigorous work. Addressing these minor points will further improve clarity and impact.
Sincerely,
Author Response
Comment 1: Streamline dense technical sections (e.g., lists of motifs, exons, or cis-elements) and consider presenting the data in clear tables either within the main manuscript or as supplementary files to improve readability and clarity.
Response 1: Thank you for this constructive suggestion. To improve readability and clarity, we have revised Figure 2 by labeling the panels as "a–d" and moved the detailed lists of cis-acting elements and motifs (Motif 1–10) to the supplementary materials. Additionally, we have updated the figure legend as follows:" Figure 2. Phylogenetic relationships, conserved motifs, distribution of cis-acting elements in the promoters, and gene structure of the TCP gene family members in C. illinoinensis, A. sinensis, and J. regia. (a) Phylogenetic analysis; (b) conserved motifs; (c) distribution of promoter cis-acting elements; and (d) gene structure. Note: in the promoter cis-acting elements section, light grey bands indicate regions with no predicted cis-acting elements, while different color blocks represent different cis-acting elements." We have also added two supplementary materials: "Figure S2: Motif logo analysis of the TCP gene family" and "Table S4: Predicted cis-acting regulatory elements in the promoter regions of TCP genes". These revisions are located on page 8, lines 253–258, and page 14, lines 415 and 418–419.
Comment 2: Presenting the research objectives more explicitly.
Response 2: Thank you for this helpful comment. Following your advice, we have revised both the Abstract and Introduction to more clearly present the research objectives. In the Abstract, the revised sentence now reads:" Here, we aimed to systematically identify, classify, and characterize TCP genes across three nut-producing Juglandaceae species—Carya illinoinensis, Annamocarya sinensis, and Juglans regia—to elucidate their evolutionary relationships and potential functions in fruit development." In the Introduction, the objectives have been revised as follows:" The main objectives of this study were to (1) comprehensively identify TCP genes in the three species, (2) analyze their phylogenetic relationships, conserved motifs, sequence alignment, chromosomal distribution, synteny, and cis-acting regulatory elements, and (3) explore their expression patterns in fruit tissues to reveal potential roles in fruit growth and development." These changes have been highlighted in yellow in the revised manuscript (page 1, lines 29–31; page 4, lines 111–115).
Comment 3: Highlighting potential practical applications or implications for crop improvement in the conclusions.
Response 3: We appreciate this insightful recommendation. In response, we have emphasized the potential practical applications and implications for crop improvement in both the Abstract and Conclusions sections. In the Abstract, the revised sentence now reads:" Collectively, these findings advance our understanding of TCP gene evolution and provide potential molecular targets for improving fruit development and nut quality in Juglandaceae crops." In the Conclusions, we have revised the text as follows:" Overall, this study provides new insights into the evolutionary history and functional specialization of TCP genes in Juglandaceae. Beyond advancing basic understanding, these findings identify promising TCP candidates for molecular breeding aimed at enhancing fruit development, yield, and quality in nut crops." These revisions have been highlighted in yellow in the manuscript (page 2, lines 47–49; page 14, lines 408–412).
